# Synthesize, Partition, then Adapt:
# Eliciting Diverse Samples from Foundation Models

**Yeming Wen**[*]**& Swarat Chaudhuri**
Department of Computer Science
The University of Texas at Austin

## Abstract

Presenting users with diverse responses from foundation models is crucial for enhancing user experience and accommodating varying preferences. However, generating multiple high-quality and diverse responses without sacrificing accuracy remains a challenge, especially when using greedy sampling. In this work, we propose a novel framework, Synthesize-Partition-Adapt (SPA), that leverages the abundant synthetic data available in many domains to elicit diverse responses from foundation models. By leveraging signal provided by data attribution methods such as influence function, SPA partitions data into subsets, each targeting unique aspects of the data, and trains multiple model adaptations optimized for these subsets. Experimental results demonstrate the effectiveness of our approach in diversifying foundation model responses while maintaining high quality, showcased through the HumanEval and MBPP tasks in the code generation domain and several tasks in the natural language understanding domain, highlighting its potential to enrich user experience across various applications.

## 1  Introduction

Transformer-based foundation models have revolutionized the fields of natural language processing (NLP) and code generation with their remarkable abilities a wide range of understanding and generation tasks (Vaswani et al., 2017; Devlin et al., 2019; Brown et al., 2020; Chen et al., 2021). These models are typically pre-trained on vast amounts of text data and then undergo instruction fine-tuning — a post-training process — to improve alignment with user expectations and enhance the overall user experience (Ouyang et al., 2022). Due to the high cost of human-annotated data, synthetically generated datasets (Wang et al., 2022b) such as OSS-Instruct (Wei et al., 2023) and Al-

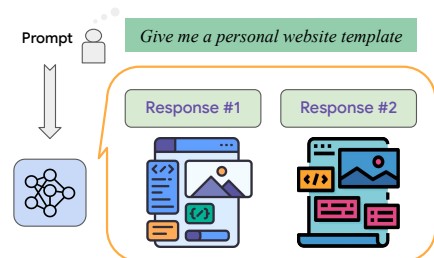

Figure 1: A user is expecting two diverse templates from the foundation model.

paca (Taori et al., 2023) have become an important component of instruction tuning, demonstrating strong effectiveness in improving foundation model performance.

To date, these synthetic datasets have been primarily used to align foundation models with instructions or to induce certain preferable behaviors. In this paper, we focus on a different use of synthetic data: in improving the *diversity* of foundation models' outputs. Diversifying the generated responses is crucial for accommodating diverse user preferences and enhancing user satisfaction. Consider the scenario illustrated in Fig. 1, where a user prompts a foundation model with "Give me a personal website template". In this case, we would prefer the model to generate two diverse templates while

38th Conference on Neural Information Processing Systems (NeurIPS 2024).

---

[*]ywen@utexas.edu

maintaining good quality, providing users with a variety of styles and layouts. Conventional methods for improving diversity, such as temperature sampling (Ackley et al., 1985; Hinton et al., 2015; Wang et al., 2019, 2023), rely on sampling techniques that anneal the probabilistic distribution of outputs. These methods often trade off diversity for quality, as the generated responses may deviate from the learned distribution and produce hallucination or less coherent outputs (Lee, 2023). Moreover, these techniques are not applicable when using greedy sampling, which is often preferred for its simplicity and precision. This highlights the need for approaches that not only align foundation model outputs with user expectations but also elicit diverse responses without sacrificing quality.

In this paper, we present a framework, *Synthesize-Partition-Adapt* (SPA), that achieves these objectives. The framework partitions the synthetic data and adapts foundation models to these partitions in the post-training stage. By leveraging the inherent diversity in the training data, this approach can generate diverse responses without compromising accuracy. The potential of partition-and-adapt approach is further amplified by the increasing availability of large-scale synthetic datasets because the utility of instruction-tuning a single model on the entire dataset diminishes. In particular, we show that influence function (Koh & Liang, 2017) can be an effective signal to partition synthetic datasets into subsets, each targeting unique aspects that elicit distinct model behaviors. However, SPA is not limited to influence function and can be extended to other partitioning strategies. By training multiple adaptations on these subsets using parameter-efficient fine-tuning techniques, such as LoRA (Hu et al., 2021), we enable the generation of diverse and accurate responses.

To demonstrate the effectiveness of our approach, we conduct experiments on a range of tasks in both the code generation and natural language understanding domains. We evaluate our method on the HumanEval (Chen et al., 2021) and MBPP (Austin et al., 2021) datasets for code generation, as well as several natural language understanding tasks. The results showcase the ability of our approach to diversify model responses while maintaining high accuracy, highlighting its potential to enrich user experience across various applications.

To summarize, the main contributions of this paper are as follows:

- We propose SPA, a novel framework that leverages synthetic data, data partitioning, and model adaptation to elicit diverse responses from foundation models.

- We demonstrate the effectiveness of SPA in diversifying foundation model responses while maintaining sampling quality through extensive experiments on code generation and natural language understanding tasks.

- We highlight the potential of SPA to leverage the increasing availability of large-scale synthetic datasets for improving the diversity of foundation model responses.

## 2 Background

### 2.1 Instruction Fine-tuning

By fine-tuning foundation models on human-annotated data that demonstrates desired behaviors, instruction tuning aims to improve the alignment between the model's outputs and the user's intentions (Ouyang et al., 2022; Wei et al., 2021; Sanh et al., 2022). Let $\mathcal{D} = (x_i, y_i)_{i=1}^{N}$ denote a dataset of input-output pairs, where $x_i$ represents the input instruction and $y_i$ represents the corresponding desired output. The objective of instruction tuning is to minimize the following loss function: $\mathcal{L}(\theta) = -\frac{1}{N} \sum_{i=1}^{N} \log_{\theta}(y_i|x_i)$ where $\theta$ represents the parameters of the foundation model, and $p_{\theta}(y_i|x_i)$ is the probability of generating the target response $y_i$ given the input $x_i$.

Classical approaches for instruction tuning typically require a substantial amount of parallel labeled data of NL intents and gold model responses. Collecting large-scale, high-quality annotated datasets is often time-consuming and expensive. To mitigate this issue, researchers have explored the use of synthetic data for instruction tuning. By leveraging techniques such as data augmentation (Wei & Zou, 2019; Sennrich et al., 2016) and back-translation (Edunov et al., 2018), synthetic data can be generated at scale, providing a cost-effective alternative to human-annotated datasets. Furthermore, synthetic instruction-following data can also be generated from the foundation model itself (Wang et al., 2022a; Honovich et al., 2022; Taori et al., 2023; Peng et al., 2023; Wen et al., 2024, *inter alia*).

## 2.2 Data Attribution and influence function

Data attribution methods aim to quantify the importance or influence of individual training points on a model's predictions. One such method is the influence function (Koh & Liang, 2017). Formally, let $\mathcal{L}(\theta)$ denote the loss function of the model, where $\theta$ represents the model parameters. The influence of a training point $z$ on the model's parameters $\theta$ is given by $\mathcal{I}(z) = -H_\theta^{-1}\nabla_\theta\mathcal{L}(z, \theta)$. where $H_\theta$ is the Hessian matrix of the loss function with respect to the model parameters, and $\nabla_\theta\mathcal{L}(z, \theta)$ is the gradient of the loss function with respect to the model parameters, evaluated at the training point $z$. Next, the influence of elevating the weight of $z$ on the loss associated with a test point $z_{test}$ is:

$$\mathcal{I}(z, z_{test}) = -\nabla_\theta\mathcal{L}(z_{test}, \hat{\theta})^\top H_{\hat{\theta}}^{-1}\nabla_\theta\mathcal{L}(z, \hat{\theta}) \qquad (1)$$

It is impossible to calculate the full Hessian $H_\theta^{-1}$ matrix in deep neural networks. Koh & Liang (2017) developed a simple and efficient implementation that requires only oracle access to gradients and Hessian-vector products. This implementation makes it feasible to apply influence function to large-scale models. However, the vast parameter space of foundation models presents an even greater challenge, rendering the direct application of influence function impractical. In response to this, recent advancements in Grosse et al. (2023) have further refined the methodology, enabling the application of influence function to large language models.

## 3 Problem Formulation

Given a user input $\mathbf{x}$, our goal is to generate a diverse set of high-quality responses $\mathbf{y}_1, \mathbf{y}_2, ..., \mathbf{y}_K$ from a foundation model $\mathcal{M}$. One approach to generating diverse responses is to sample from the model multiple times using techniques like temperature sampling: $\mathbf{y}_k = \mathcal{M}(\mathbf{x}; \theta, \tau)$, where $k = 1, 2, ..., K$ and $\theta$ represents the model parameters and $\tau$ is the temperature hyperparameter. However, this approach often trades off diversity for quality as studied in Chung et al. (2023). An alternative approach is to train multiple model adaptations $\mathcal{M}_1, \mathcal{M}_2, ..., \mathcal{M}_K$ and sample one response from each adaptation:

$$\mathbf{y}_k = \mathcal{M}_k(\mathbf{x}; \theta_k), \quad k = 1, 2, ..., K, \qquad (2)$$

where $\theta_k$ represents the parameters of the $k$-th model adaptation. By training each adaptation on a different subset of the data that captures unique aspects and yields distinct model behaviors, we can generate diverse responses while maintaining their quality. Moreover, this approach allows us to elicit diverse samples even with greedy sampling, which is often preferred for maximum precision.

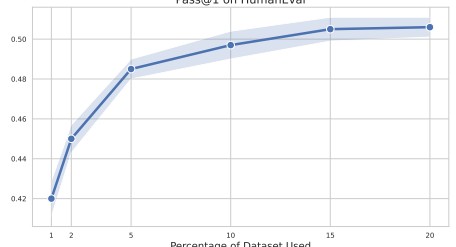

Figure 2: *pass@1* on HumanEval after fine-tuning on some percentage of OSS-Instruct dataset (Wei et al., 2023) using LoRA. The plot demonstrates the diminishing returns observed with increasing amounts of data used for parameter efficient fine-tuning.

Traditionally, training multiple model adaptations has been considered unfavorable due to the repeated training process, which can be computationally expensive and time-consuming. However, with the increasing popularity of instruction tuning, it has become common practice to go through a post-training stage using instruction data before deploying the model to users. This post-training stage presents an opportunity to train multiple model adaptations without incurring significant additional costs, making the approach more feasible and practical in real-world scenarios.

As the volume of synthetic data grows, the utility of fine-tuning a single model on the entire dataset diminishes due to the diminishing returns in the post-training stage, as demonstrated in Fig. 2. The *pass@1* accuracy after fine-tuning on the entire synthetic dataset using LoRA is roughly the same as only consuming 15% of the data[2]. This creates an opportunity to leverage the abundant synthetic data to train multiple model adaptations, each specializing in a specific subset of the data. In this work, we propose the Synthesize, Partition, then Adapt (SPA) framework to address the diverse response generation problem. SPA leverages existing synthetic datasets, data partitioning techniques, and parameter-efficient fine-tuning methods to train multiple model adaptations. By sampling from the collection of these adaptations, SPA generates diverse and high-quality responses, enhancing the overall user experience.

---

[2]This does not suggest full parameter fine-tuning shares the same diminishing return.

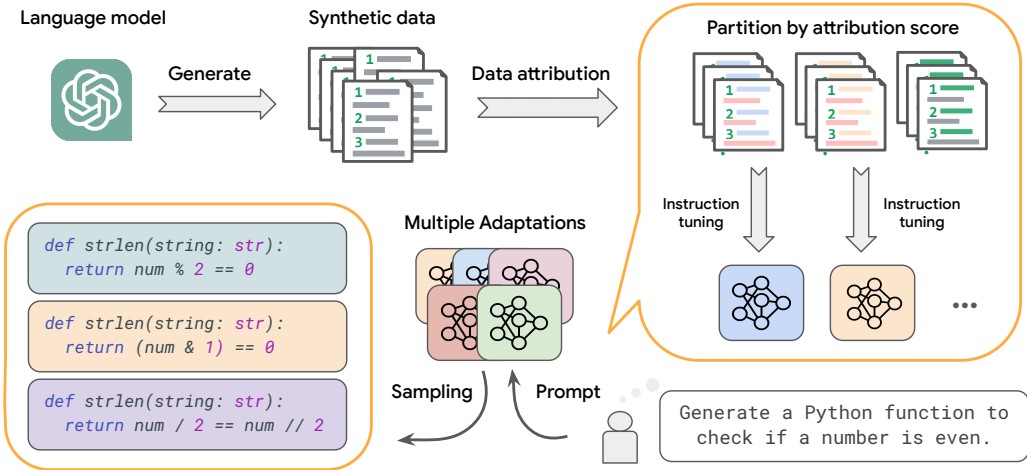

Figure 3: An illustration of the Synthesize, Partition, then Adapt (SPA) framework. SPA partitions synthetic dataset according to data attribution scores, which can be obtained using various methods such as influence function or lexical overlap. Multiple foundation model adaptations are then trained on each subset. Sampling from the collection of these model adaptations can present users with diverse responses. SPA is not limited to a specific attribution method.

## 4   Partitioning Synthetic Data and Training Adaptations

We present the technical details of our proposed SPA framework for training multiple adaptations. We leverage an existing synthetic dataset $\mathcal{D} = \{\mathbf{x}_i, \mathbf{y}_i\}_{i=1}^{N}$ for the purpose of this study. The use of an existing synthetic dataset allows us to focus on the effectiveness of the Partition then Adapt steps in eliciting diverse samples, while demonstrating the flexibility of our framework to work with various synthetic datasets. Fig. 3 provides an overview of the framework. After obtaining the synthetic data, our approach consists of three main steps: (1) computing data attribution scores for synthetic data points, (2) partitioning the synthetic dataset based on these scores, and (3) training multiple foundation model adaptations using parameter-efficient fine-tuning techniques like LoRA.

### 4.1   Computing Data Attribution Scores

Consider a pre-trained foundation model $\mathcal{M}$ with parameters $\theta$. Our goal is to leverage the synthetic dataset $\mathcal{D}$ to train a set of $K$ foundation model adaptations $\{\mathcal{M}_k\}_{k=1}^{K}$. Each adaptation focuses on a specific subset of the data that yields similar model behaviors. To partition the synthetic dataset, we employ data attribution methods that measure the importance of each training point to the model's predictions. Although we use influence function as an example to label the data, the SPA framework is not limited to influence function and can be extended to other data attribution methods, such as lexical overlap or TRAK (Park et al., 2023). To calculate the influence function, we first fine-tune the pre-trained foundation model $\mathcal{M}$ on the synthetic dataset $\mathcal{D}$. The fine-tuning process optimizes the model parameters $\theta$ to minimize the loss function $\mathcal{L}(\theta)$ on the synthetic dataset using LoRA (Hu et al., 2021): $\mathcal{L}(\theta) = \frac{1}{N} \sum_{i=1}^{N} \ell(\mathbf{y}_i, \mathcal{M}(\mathbf{x}_i; \theta))$, where $\ell(\cdot, \cdot)$ is a suitable loss function, such as cross-entropy loss for language modeling tasks. This fine-tuning process yields the optimized model parameters $\hat{\theta}$.

Next, we select a set of $M$ test queries $\{(\mathbf{x}_t^{(m)}, \mathbf{y}_t^{(m)})\}_{m=1}^{M}$, which can be a collection of questions requiring various expertise knowledge to solve. For each test query $(\mathbf{x}_t^{(m)}, \mathbf{y}_t^{(m)})$, we compute the influence score of each synthetic data point $(\mathbf{x}_i, \mathbf{y}_i) \in \mathcal{D}$ using Eq. (1):

$$\mathcal{I}((\mathbf{x}_i, \mathbf{y}_i), (\mathbf{x}_t^{(m)}, \mathbf{y}_t^{(m)})) = -\nabla_\theta \ell(\mathbf{y}_t^{(m)}, \mathcal{M}(\mathbf{x}_t^{(m)}; \hat{\theta}))^\top H_{\hat{\theta}}^{-1} \nabla_\theta \ell(\mathbf{y}_i, \mathcal{M}(\mathbf{x}_i; \hat{\theta})). \qquad (3)$$

To efficiently compute the influence scores, we employ the stochastic estimation method proposed by Koh & Liang (2017), which approximates the inverse Hessian-vector product using conjugate

gradients. Although even this method is generally infeasible in foundation models due to their vast parameter space, the use of LoRA (Hu et al., 2021) makes it feasible by significantly reducing the number of trainable parameters. The computational cost of estimating the influence of a test query between the entire dataset $\mathcal{D}$ is the same as calculating the gradient of $\mathcal{D}$. Another option to address this issue is to use the K-FAC approximation of the Hessian, as proposed by Grosse et al. (2023). We focus on the LoRA approach and leave the exploration of K-FAC and other approximations for future work.

## 4.2 Partitioning Synthetic Dataset

After computing the data attribution scores for each synthetic data point with respect to the $M$ test points, we obtain an influence matrix $\mathbf{I} \in \mathbb{R}^{N \times M}$, where $\mathbf{I}_{i,m}$ represents the attribution score of the $i$-th synthetic data point for the $m$-th test point. To partition the synthetic dataset $\mathcal{D}$ into $K$ subsets $\{\mathcal{D}_k\}_{k=1}^K$, a clustering algorithm can be applied to solve the following objective:

$$\min_{\{\mathcal{D}_k\}_{k=1}^K} \sum_{k=1}^K \sum_{(\mathbf{x}_i, \mathbf{y}_i) \in \mathcal{D}_k} \sum_{(\mathbf{x}_j, \mathbf{y}_j) \in \mathcal{D}_k} |\mathbf{I}_{i,:} - \mathbf{I}_{j,:}|_2^2, \tag{4}$$

where $\mathbf{I}_{i,:}$ denotes the $i$-th row of the influence matrix $\mathbf{I}$, subject to $\bigcup_{k=1}^K \mathcal{D}_k = \mathcal{D}$ and $\mathcal{D}_k \cap \mathcal{D}_{k'} = \emptyset$ for all $k \neq k'$. In this work, we assume partitions are disjoint for the simplicity of the study.

The clustering algorithm assigns each synthetic data point $(\mathbf{x}_i, \mathbf{y}_i)$ to one of the $K$ subsets based on the similarity of its influence scores across the $M$ test points. This partitioning ensures that data points within each subset have similar impacts on the model's predictions. The choice of the clustering algorithm may depend on the specific characteristics of the dataset. For simplicity and ease of implementation, in this study, we use a ranking heuristic to partition the synthetic dataset. The details of this heuristic will be explained in the experiment section §5.1. However, it is important to note that our SPA framework is not limited to any specific clustering algorithm.

## 4.3 Training Multiple Adaptations with LoRA

Once the synthetic dataset is partitioned into $K$ subsets, we train a foundation model $\mathcal{M}_k$ for each subset $\mathcal{D}_k$ using parameter-efficient fine-tuning techniques like LoRA (Hu et al., 2021). LoRA adapts the pre-trained foundation model parameters $\theta$ by learning low-rank matrices $\mathbf{A}_k \in \mathbb{R}^{r \times d}$ and $\mathbf{B}_k \in \mathbb{R}^{d \times r}$ for each weight matrix $\mathbf{W} \in \mathbb{R}^{d \times d}$ in the pre-trained foundation model, where $r \ll d$ is the rank of the adaptation matrices.

The adapted weight matrix $\mathbf{W}_k$ for the foundation model adaptation $\mathcal{M}_k$ is computed as: $\mathbf{W}_k = \mathbf{W} + \mathbf{B}_k \mathbf{A}_k$. During the fine-tuning process, only the adaptation matrices $\mathbf{A}_k$ and $\mathbf{B}_k$ are learned, while the pre-trained weights $\mathbf{W}$ remain frozen. This significantly reduces the number of trainable parameters, making it feasible to train multiple foundation model adaptations with limited computational resources. The training objective for each foundation model adaptation $\mathcal{M}_k$ is given by $\min_{\theta_k} \frac{1}{|\mathcal{D}_k|} \sum_{(\mathbf{x}_i, \mathbf{y}_i) \in \mathcal{D}_k} \ell(\mathbf{y}_i, \mathcal{M}_k(\mathbf{x}_i; \theta_k))$ where $\theta_k$ represents the parameters of $\mathcal{M}_k$, which include the pre-trained weights $\theta$ and the LoRA adaptation matrices $\mathbf{A}_k, \mathbf{B}_k$. By training multiple foundation model adaptations using LoRA, we can efficiently adapt the pre-trained foundation model to different subsets of the synthetic data, each focusing on a specific aspect of the data that yields similar model behaviors. This approach enables the creation of a diverse set of specialized models that capture different knowledge or expertise present in the synthetic data, while leveraging the knowledge acquired during the pre-training phase.

**Inference with Multiple Adaptations**   During inference, given a user input $\mathbf{x}$, our goal is to generate a diverse set of responses by leveraging the multiple foundation model adaptations trained on different subsets of the synthetic data. To achieve this, we randomly sample a foundation model adaptation $\mathcal{M}_k$ from the set of $K$ adaptations $\{\mathcal{M}_k\}_{k=1}^K$ and generate the output $\mathbf{y}$ using the selected adaptation. By randomly sampling from the set of adaptations, we can generate a diverse set of responses for the user input $\mathbf{x}$. This approach ensures that the generated responses are not only diverse but also maintain reasonable quality. It is worth noting that this approach is compatible with various sampling techniques, such as temperature scaling, top-k and top-p sampling, which can further enhance the diversity of the generated responses.

To generate multiple diverse responses for the user input $\mathbf{x}$, we can repeat the random sampling process multiple times, each time selecting a different adaptation and generating a response. This allows us to present the user with a set of alternative responses that capture different perspectives or styles, enhancing the overall user experience. Unlike temperature sampling, which can degrade the quality of the generated responses, our approach maintains the quality of each response by leveraging the specialized knowledge captured by each adaptation. Moreover, our approach can generate diverse samples even when greedy sampling is used.

## 5 Experiments

In this section, we present the experimental setup and results for evaluating the effectiveness of our proposed SPA framework in improving the diversity of foundation model outputs. We conduct experiments on both code generation tasks such as HumanEval (Chen et al., 2021) and MBPP (Austin et al., 2021) and several natural language understanding tasks.

### 5.1 Experimental Setup

**Base Model and Synthetic Dataset**   For the code generation experiments, we use CodeLLaMA 7B (Rozière et al., 2023) as the base foundation model. CodeLLaMA is a state-of-the-art language model specifically designed for code-related tasks, pre-trained on a large corpus of code and natural language data. For the synthetic dataset, we utilize the OSS-Instruct dataset (Wei et al., 2023), which consists of 75,000 code-related question-answering pairs generated by GPT-3.5 Turbo (OpenAI, 2023). In the natural language understanding domain, we employ Llama-2 13B (Touvron et al., 2023) as the base foundation model. Llama-2 is a powerful language model trained on a diverse range of web-scale data, demonstrating strong performance across various natural language understanding tasks. For the synthetic dataset, we use Platypus (Lee et al., 2023), which focuses on improving LLMs' STEM and logic knowledge. Platypus consists of a curated sub-selection of public text datasets, comprising approximately 25,000 question-answer pairs.

**Data Attribution Scores**   We compare two methods for computing data attribution scores: influence function and lexical overlap.

For the influence-based method, we hand-write 12 examples that cover a wide range of knowledge for each domain. For each of these examples, we calculate the influence score with respect to each training example in the corresponding synthetic dataset using Equation 3. We then select the top 8 test queries whose distribution of influence scores over the dataset has the highest variance. This ensures that the selected test queries have diverse impacts on the synthetic dataset, capturing different aspects of the domain knowledge. The resulting influence matrices $\mathbf{I}_{code} \in \mathbb{R}^{8 \times 75,000}$ and $\mathbf{I}_{nlu} \in \mathbb{R}^{8 \times 25,000}$ are used for partitioning the OSS-Instruct and Platypus datasets, respectively.

For the lexical overlap method, we compute the BM25 score (Robertson et al., 1994) between each training example and the hand-written test queries. The BM25 score is calculated as follows:

$$I(z, z_{query}) = \sum_{t \in z_{query}} \log \frac{N+1}{N_t} \cdot \left( \frac{(k_1+1)f(z,t)}{k_1\left((1-b) + b \cdot \frac{L(z)}{L_{avg}}\right) + f(z,t)} + 1 \right) \quad (5)$$

where $f(z,t)$ is the overlap count, $N$ is the number of training examples, $L(z)$ is the length of the example, and $L_{avg}$ is the average example length. We adopted the framework and the hyperparameters in Lv & Zhai (2011). While we focus on influence function in this work, exploring the effectiveness of alternative data attribution methods like BM25 could be an interesting direction for future research. More details are provided in Appendix A.

**Partitioning the Synthetic Datasets**   To train multiple foundation model adaptations, we first set the hyperparameter $K$, which represents the total number of adaptations. We use $K = 8$ for both code generation and natural langauge understanding domain. For each data point in the synthetic dataset, we aim to find the test queries that provides the most influence. Formally, for each synthetic data point $(\mathbf{x}_i, \mathbf{y}_i)$, we assign it to the subset $\mathcal{D}_k^*$ corresponding to the test point with the highest influence score or the BM25 score: $k^* = \arg\max_{k \in \{1, \ldots, K\}} \mathbf{I}_{k,i}$. where $\mathbf{I}_{k,i}$ represents either the influence matrix or the BM25 score matrix. This process partitions the OSS-Instruct dataset

| Methods | HUMANEVAL | | | | MBPP | | | |
|---|---|---|---|---|---|---|---|---|
| | $pass@1$ | $pass@5$ | diversity | avg. KL | $pass@1$ | $pass@5$ | diversity | avg. KL |
| Single ($\tau = 0.1$) | 50.02 | 56.42 | 0.58 | NA | 60.15 | 64.16 | 0.53 | NA |
| Random ($\tau = 0$) | 50.15 | 63.10 | 0.69 | 0.008 | **60.65** | 70.42 | 0.64 | 0.014 |
| Lexical ($\tau = 0$) | **50.30** | 66.74 | 0.78 | 0.011 | 60.33 | 71.17 | 0.71 | 0.018 |
| Influence ($\tau = 0$) | 50.15 | **69.05** | **0.85** | **0.017** | 60.46 | **73.68** | **0.78** | **0.020** |

Table 1: Results on the HumanEval and MBPP. $\tau$ denotes the temperature used for sampling. SPA with influence function achieves the best performance in terms of diversity score and avg. KL divergence) while maintaining comparable $pass@1$ performance to the single adaptation baseline. $pass@5$ measures sample quality but also has a positive correlation with diversity.

into $K$ groups for code generation and the Platypus dataset into $K$ groups for natural language understanding. Each group is associated with a specific test example that has the highest influence on the data points within the group.

With the partitioned synthetic dataset, we train $K$ model adaptations using the LoRA technique, as described in §4.3. Each adaptation $\mathcal{M}_k$ is trained on the corresponding subset $\mathcal{D}_k$ of the synthetic dataset, focusing on the specific coding knowledge captured by the associated test point.

**Evaluation Metrics**   We use the following two metrics to assess the diversity:

1. Average KL Divergence: Let $P_i$ and $P_j$ be the probability distributions of the generated responses from two model adaptations $i$ and $j$, respectively. The KL divergence between $P_i$ and $P_j$ is defined as $D_{KL}(P_i \parallel P_j) = \sum_x P_i(x) \log \frac{P_i(x)}{P_j(x)}$. The average KL divergence is calculated by averaging the pairwise KL divergence between all possible pairs of model adaptations. A higher average KL divergence indicates greater diversity among the model adaptations,

$$\text{Average KL Divergence} = \frac{1}{\binom{K}{2}} \sum_{i=1}^{N-1} \sum_{j=i+1}^{N} D_{KL}(P_i \parallel P_j) \tag{6}$$

2. Sample Diversity: The average KL divergence evaluates the diversity at the distributional level. We also consider the sample diversity which measures the uniqueness of individual responses. We calculate the diversity score among $K$ randomly generated samples for each problem. The diversity score is defined as the proportion of unique samples within the generated set. Specifically, it is calculated by taking one minus the ratio of the number of duplicate pairs to the total number of generated pairs.

**Baselines**   We consider two baselines in the evaluation: (1) **Single Adaptation**, where a single model adaptation is trained on the entire synthetic dataset using LoRA, and (2) **Multiple Adaptations (random)**, where multiple adaptations are trained on randomly partitioned subsets of the synthetic dataset using LoRA. Hyperparameters used to train adaptations are provided in Appendix A.

## 5.2   Code Generation Results

In the code generation domain, we evaluate the performance of our proposed methodology on two popular code generation benchmarks: HumanEval (Chen et al., 2021) and MBPP (Austin et al., 2021). HumanEval consists of 164 hand-written programming problems with corresponding test cases, while MBPP contains 399 held-out programming problems collected from online resources[3]. These benchmarks assess the ability to generate functionally correct code.

**$pass@k$ metric** In addition to the diversity metrics, we also evaluate the sample quality by $pass@1$ and $pass@5$, measuring the percentage of problems for which at least one of the $k$ generated samples passes all the test cases. Note that the $pass@5$ metric has a strong correlation to the diversity of the samples. More diverse samples generally lead to higher $pass@5$ for $k > 1$.

Tab. 1 presents the evaluation results of our SPA framework and the baselines on the HumanEval and MBPP benchmarks. For the multiple adaptation methods, including random partitioning, lexical

---

[3]We used the evalplus (Liu et al., 2023) framework to evaluate samples.

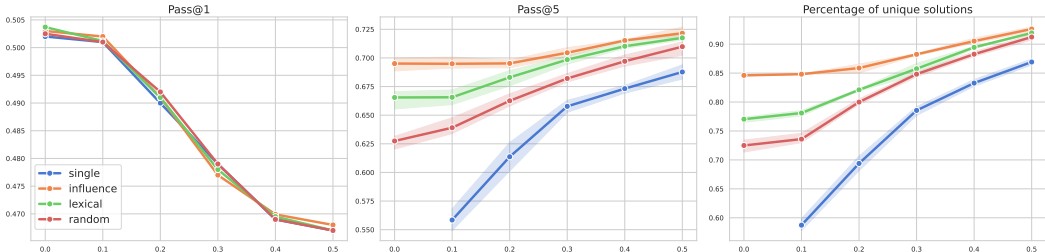

Figure 4: How sampling temperature affects $pass@1$, $pass@5$, and Diversity Score for different methods on the HumanEval benchmark. The results are averaged over 4 checkpoints.

overlap, and influence function, we use greedy decoding ($\tau = 0$) to generate samples. For the single adaptation baseline, we use a temperature of $\tau = 0.1$ to induce some diversity in the generated samples, as greedy decoding would not produce any diversity in this case.

Our primary focus is on comparing the diversity metrics, namely the Diversity Score and the Average KL Divergence (avg. KL), across the different methods. SPA with influence function achieves the highest Diversity Scores of 85% and 78% on HumanEval and MBPP, respectively, indicating that the generated samples are more unique and diverse compared to the other methods. Similarly, SPA with influence function yields the highest Average KL Divergence of 0.017 and 0.020 on HumanEval and MBPP, demonstrating greater diversity at the distributional level.

The random partitioning and lexical overlap approaches also improve upon the single adaptation baseline in terms of diversity metrics, but to a lesser extent than influence function. In particular, the lexical overlap induces more diversity than the random adaptations baseline. This suggests that even simpler data attribution methods can be beneficial for enhancing diversity when training multiple specialized adaptations. It is worth noting that the $pass@5$ scores, while primarily measuring sample quality, also have a positive correlation with diversity. SPA with influence function achieves the highest $pass@5$ scores of 69.05% and 73.68% on HumanEval and MBPP, indicating that the generated samples not only exhibit greater diversity but also maintain high quality.

In summary, these results underscore the effectiveness of our SPA framework in generating diverse code samples without compromising quality. By leveraging influence function for data partitioning and training multiple adaptations using LoRA, SPA enables the generation of diverse and accurate code solutions, even when using greedy decoding. We also showed that training more adaptations than 8 did not lead to more diversity in Appendix B.

**Impact of Temperature**   Fig. 4 presents the impact of temperature on $pass@1$, $pass@5$, and Diversity Score for different methods on the HumanEval benchmark. The first plot shows that all methods, including Single, Random, Lexical, and Influence, exhibit similar patterns in terms of $pass@1$ performance. They achieve maximum accuracy (around 50.2%) when $\tau = 0$ and gradually decrease to approximately 46.5% as the temperature increases to 0.5.

However, both $pass@5$ and Diversity Score improve for all methods as the temperature increases, which is expected as higher temperatures encourage the model to generate more diverse samples. Notably, SPA with influence function (Influence) maintains its advantage over other methods across all temperature values, outperforming Single, Random, and Lexical methods. Although the performance gap between Influence and other methods narrows as the temperature increases due to the inherent diversity promotion of higher temperatures, Influence still maintains a lead at $\tau = 0.5$.

### 5.3   Natural Language Understanding Results

To demonstrate the effectiveness of SPA in the natural language understanding domain, we evaluate its performance on several diverse tasks, including Big-Bench Hard (BBH) (Suzgun et al., 2022), GPQA (Rein et al., 2023), MMLU (Hendrycks et al., 2020), and WinoGrande (Sakaguchi et al., 2019). For tasks that involve multiple-choice questions, we asked the model to continue generating text even after producing an answer choice for the purpose of measuring sample diversity. As shown in Fig. 5, SPA with influence function consistently achieves higher diversity scores and average KL divergence compared to the lexical overlap and random adaptation across all tasks. Interestingly,

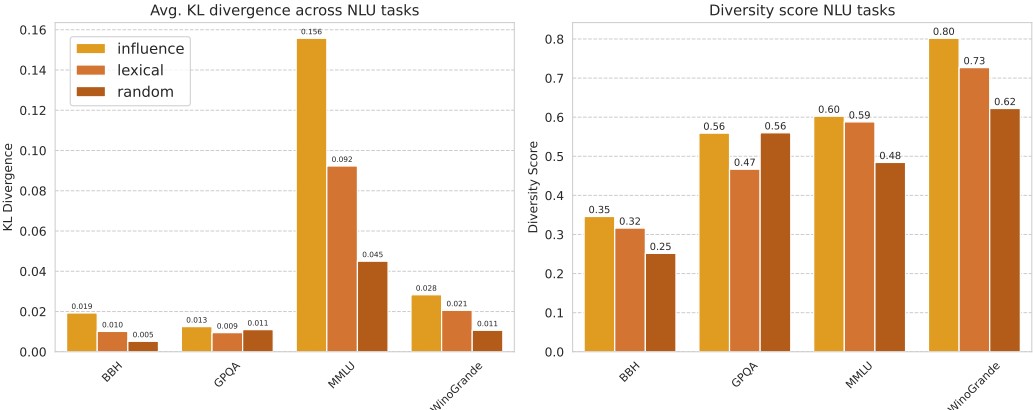

Figure 5: Average KL divergence and diversity score on various natural language understanding tasks. SPA with influence function consistently outperforms the lexical overlap and random adaptations, demonstrating its effectiveness in generating diverse samples across different NLU tasks.

random adaptations achieve better diversity than lexical overlap on the GPQA task, suggesting that the effectiveness of partitioning methods may change depending on the task.

The diversity scores and average KL divergence values vary across tasks, reflecting the inherent differences in the nature and complexity of each task. Tasks like MMLU, which cover a wide range of subjects, tend to yield higher average KL divergence. We also notice that a larger gap in average KL divergence does not necessarily translate to a proportionally greater difference in diversity scores. This suggests that while average KL divergence captures the dissimilarity between the generated sample distributions, it may not always directly correlate with the actual diversity of the samples. Nonetheless, the consistent improvement achieved by SPA with influence function highlights its robustness and adaptability to various natural language understanding challenges.

# 6   Related Work

Sampling-based methods have been widely explored to generate diverse text from language models. One of the most common approaches is temperature sampling (Ackley et al., 1985; Hinton et al., 2015). Several studies have investigated the impact of temperature on model sampling and its effect on the diversity-quality trade-off (Caccia et al., 2018; Renze & Guven, 2024; Wang et al., 2023). Higher temperatures lead to more diverse but potentially less coherent samples, while lower temperatures produce more conservative and deterministic outputs. When using high temperatures, human interventions can help to correct errors during the sampling process (Chung et al., 2023). Dynamic temperature strategies have also been explored during the model training and inference stages (Lin et al., 2018; Zhang et al., 2018; Wang et al., 2019; Chang et al., 2023).

Besides adjusting temperature, top-$k$, top-$p$ (nucleus) sampling (Holtzman et al., 2019) and their variants are common sampling methods (Fan et al., 2018; Meister et al., 2022; Hewitt et al., 2022; Ravfogel et al., 2023), which restrict the sampling space or dynamically adjust the number of tokens considered at each step. Another line of works studied how to formulate quality-diversity trade-off as a search or RL problem (Naik et al., 2023; Lim et al., 2024; Mudgal et al., 2023; Bradley et al., 2023; Ji et al., 2023).

# 7   Conclusion

In summary, we proposed SPA, which that leverages synthetic data, data partitioning, and model adaptation to elicit diverse responses from foundation models. By partitioning synthetic datasets into subsets that capture unique aspects of the data and training multiple model adaptations optimized for these subsets, SPA enables the generation of diverse and high-quality responses.

**Limitation**   One main challenges is the computational cost associated with influence function, which require several extra epochs of backward passes to estimate. Future work could explore more efficient data attribution methods, such as TRAK (Park et al., 2023) and K-FAC (Grosse et al., 2023).

Additionally, the ranking heuristics used to approximate Eq. (4) can be replaced by more advanced clustering algorithms. Additionally, serving multiple LoRA adaptations poses significant computational challenge in real-time serving framework. Recent works such as S-LoRA and FLoRA (Sheng et al., 2023; Wen & Chaudhuri, 2024) can be considered to accommodate this overhead.

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

# A  Experimental Setup Details

This section provides additional details on the experimental setup that were not included in the main content due to space constraints.

**Computing Data Attribution Scores**   For the lexical overlap method, we use a publicly available BM25 (Lv & Zhai, 2011; Trotman et al., 2014) implementation written in Python and released under https://pypi.org/project/rank-bm25/. We used the default hyperparameters.

When calculating the influence function, we employ the conjugate gradient method with LiSSA approximation (Martens, 2010; Agarwal et al., 2016). We leverage a publicly available implementation from https://github.com/alstonlo/torch-influence/. For the OSS-Instruct dataset, we use a damping factor of 0.001, a depth of 120, and 500 repeats, following the guideline that the product of depth and repeats should be roughly equal to the dataset size. For the Platypus dataset, we use a depth of 120 and 200 repeats. It is worth noting that computing the influence function is also intensive with LoRA. Each column of the $\mathbf{I}$ matrix in Eq. (4) requires approximately one epoch of backward passes over the entire synthetic dataset. On the OSS-Instruct dataset, this takes roughly 5 hours using a single A100 80GB GPU. However, this is offline computation which is consumed before deploying the model to users.

After obtaining the data attribution matrix, we observe that using the ranking heuristic presented in §5.1 leads to imbalanced partitions. To achieve more balanced partitions, we normalize the data attribution matrix before applying the heuristics. We leave the exploration of more advanced clustering algorithms, such as k-means, for future work.

**Details for Model Adaptations**   In this section, we provide details on the computing resources and hyperparameters used for training the model adaptations in both the code generation and natural language understanding domains. For the code generation experiments, we use a machine with 3 A100 40GB GPUs and train each partition for 400 steps, which takes approximately 80 minutes (each partition). The hyperparameters are mostly adopted from `https://github.com/bigcode-project/starcoder/tree/main`. The base model is CodeLLaMA-7B-Python, and we use bf16 precision to accelerate training. The per-device train batch size is set to 1, with a gradient accumulation step of 20. We use a learning rate of 2e-4 with a cosine learning rate scheduler and 20 warmup steps. For the LORA hyperparameters, we use a rank ($r$) of 16, an alpha of 16.

In the natural language understanding domain, we train each partition for 400 steps, which takes approximately 40 minutes, using the Llama-2 13B model as the base model. The training time is shorter compared to the code generation domain because the Platypus dataset is much smaller than OSS-Instruct. The hyperparameters are mostly adopted from `https://github.com/arielnlee/Platypus`. We use a per-device batch size of 1 and a gradient accumulation step of 4. The learning rate is set to 1e-4, with a total of 20 warmup steps.

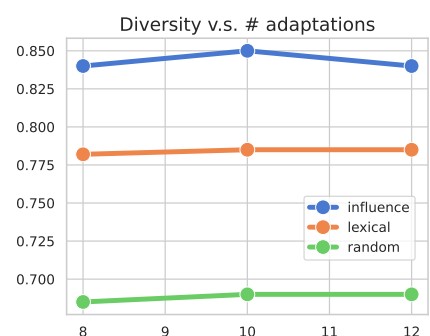

Figure 6: Diversity score as function of the number of adaptations on the HumanEval benchmark.

All the computational costs mentioned in this section, including the time and resources required for computing data attribution scores and training model adaptations, are offline. These costs are incurred before deploying the models to users, and they do not affect the inference time.

## B   Impact of Number of Adaptations

In this section, we investigate the impact of the number of model adaptations on the diversity of the generated responses. We focus on the HumanEval benchmark in the code generation domain and vary the number of adaptations from 8 to 12. The results are presented in Fig. 6.

As shown in Fig. 6, the diversity score remains relatively stable as the number of adaptations increases from 8 to 12, regardless of the partitioning method used. These results suggest that increasing the number of adaptations beyond a certain point may not necessarily lead to an improvement in the diversity of the generated responses.

## C   Test Queries

In this appendix, we provide the hand-written test queries used in our experiments for both the code generation and text generation domains. These examples were utilized to compute data attribution scores. Most of the examples are generated by GPT-4 (OpenAI, 2023).

### C.1   Code Generation Domain

```
1  1. """Title: Longest Palindromic Subsequence
2  Query: Write a function to find the longest palindromic subsequence in a given
   ↪  string.
3  Solution:
4  """
5  def longest_palindromic_subsequence(s):
6      n = len(s)
7      dp = [[0] * n for _ in range(n)]
```

```
 8
 9      for i in range(n):
10          dp[i][i] = 1
11
12      for length in range(2, n+1):
13          for i in range(n-length+1):
14              j = i + length - 1
15              if s[i] == s[j] and length == 2:
16                  dp[i][j] = 2
17              elif s[i] == s[j]:
18                  dp[i][j] = dp[i+1][j-1] + 2
19              else:
20                  dp[i][j] = max(dp[i+1][j], dp[i][j-1])
21
22      return dp[0][n-1]
23
24  2. """Title: Nth Fibonacci Number
25  Query: Implement a function to calculate the nth Fibonacci number using dynamic
    ↪  programming.
26  Solution:
27  """
28  def fibonacci(n):
29      if n <= 0:
30          return 0
31      elif n == 1:
32          return 1
33
34      fib = [0] * (n + 1)
35      fib[1] = 1
36
37      for i in range(2, n + 1):
38          fib[i] = fib[i - 1] + fib[i - 2]
39
40      return fib[n]
41
42  3. """Title: Sum of Two Largest Elements
43  Query: Create a function that takes a list of integers and returns the sum of the
    ↪  two largest elements in the list.
44  Solution:
45  """
46  def sum_of_two_largest(nums):
47      if len(nums) < 2:
48          return sum(nums)
49
50      largest = second_largest = float('-inf')
51
52      for num in nums:
53          if num > largest:
54              second_largest = largest
55              largest = num
56          elif num > second_largest:
57              second_largest = num
58
59      return largest + second_largest
60
61  4. """Title: Maximum Subarray Sum
62  Query: Implement a function to find the maximum subarray sum in a given array of
    ↪  integers.
63  Solution:
64  """
65  def max_subarray_sum(nums):
66      max_sum = float('-inf')
67      current_sum = 0
68
69      for num in nums:
```

```python
70          current_sum = max(num, current_sum + num)
71          max_sum = max(max_sum, current_sum)
72
73      return max_sum
74
75  5. """Title: First Non-Repeating Character
76  Query: Create a function that takes a string and returns the first non-repeating
    ↪  character in the string.
77  Solution:
78  """
79  def first_non_repeating_character(s):
80      char_count = {}
81
82      for char in s:
83          char_count[char] = char_count.get(char, 0) + 1
84
85      for char in s:
86          if char_count[char] == 1:
87              return char
88
89      return None
90
91  6. """Title: Merge Two Sorted Lists
92  Query: Write a function to merge two sorted lists into a single sorted list.
93  Solution:
94  """
95  def merge_sorted_lists(list1, list2):
96      merged_list = []
97      i = j = 0
98
99      while i < len(list1) and j < len(list2):
100         if list1[i] <= list2[j]:
101             merged_list.append(list1[i])
102             i += 1
103         else:
104             merged_list.append(list2[j])
105             j += 1
106
107     while i < len(list1):
108         merged_list.append(list1[i])
109         i += 1
110
111     while j < len(list2):
112         merged_list.append(list2[j])
113         j += 1
114
115     return merged_list
116
117 7. """Title: Remove Prime Numbers from List
118 Query: Create a function that takes a list of integers and returns a new list with
    ↪  all the prime numbers removed.
119 Solution:
120 """
121 def is_prime(num):
122     if num < 2:
123         return False
124     for i in range(2, int(num ** 0.5) + 1):
125         if num % i == 0:
126             return False
127     return True
128
129 def remove_prime_numbers(nums):
130     return [num for num in nums if not is_prime(num)]
131
132 8. """Title: Longest Common Substring
```

```
133   Query: Write a function to find the longest common substring between two given
      ↪  strings.
134   Solution:
135   """
136   def longest_common_substring(str1, str2):
137       m, n = len(str1), len(str2)
138       dp = [[0] * (n + 1) for _ in range(m + 1)]
139       max_length = 0
140       end_index = 0
141
142       for i in range(1, m + 1):
143           for j in range(1, n + 1):
144               if str1[i - 1] == str2[j - 1]:
145                   dp[i][j] = dp[i - 1][j - 1] + 1
146                   if dp[i][j] > max_length:
147                       max_length = dp[i][j]
148                       end_index = i
149               else:
150                   dp[i][j] = 0
151
152       start_index = end_index - max_length
153       return str1[start_index : end_index]
154
155   9. """Title: Kth Largest Element in an Unsorted Array
156   Query: Implement a function to find the kth largest element in an unsorted array.
157   Solution:
158   """
159   def kth_largest_element(nums, k):
160       k = len(nums) - k
161
162       def partition(left, right):
163           pivot = nums[right]
164           i = left - 1
165
166           for j in range(left, right):
167               if nums[j] <= pivot:
168                   i += 1
169                   nums[i], nums[j] = nums[j], nums[i]
170
171           nums[i + 1], nums[right] = nums[right], nums[i + 1]
172           return i + 1
173
174       def quick_select(left, right):
175           if left == right:
176               return nums[left]
177
178           pivot_index = partition(left, right)
179
180           if k == pivot_index:
181               return nums[k]
182           elif k < pivot_index:
183               return quick_select(left, pivot_index - 1)
184           else:
185               return quick_select(pivot_index + 1, right)
186
187       return quick_select(0, len(nums) - 1)
188
189   10. """Title: Product of Array Elements
190   Query: Create a function that takes a list of integers and returns the product of
      ↪  all the elements.
191   Solution:
192   """
193   def product_of_elements(nums):
194       product = 1
195       for num in nums:
```

```
196         product *= num
197     return product
198
199  11. """Title: Binary Search
200  Query: Implement a function to perform binary search on a sorted list of integers.
201  Solution:
202  """
203  def binary_search(nums, target):
204      left = 0
205      right = len(nums) - 1
206
207      while left <= right:
208          mid = (left + right) // 2
209
210          if nums[mid] == target:
211              return mid
212          elif nums[mid] < target:
213              left = mid + 1
214          else:
215              right = mid - 1
216
217      return -1
218
219  12. """Title: Find Missing Number
220  Query: Create a function that takes a list of integers from 0 to n (inclusive) with
       ↪  one number missing and returns the missing number.
221  Solution:
222  """
223  def find_missing_number(nums):
224      n = len(nums)
225      expected_sum = (n * (n + 1)) // 2
226      actual_sum = sum(nums)
227      return expected_sum - actual_sum
```

## C.2   Text Generation Domain

```
1   1. Title: Economic Impacts of the Black Death
2   Query: Explain the economic impacts of the Great Mortality in medieval Europe.
3   Response: The the Great Mortality drastically reduced the population of Europe,
     ↪  leading to severe labor shortages, higher wages, lower prices for land, and a
     ↪  shift in economic power from the feudal lords to the working class and
     ↪  merchants.

4
5   2. Title: Photosynthesis Process
6   Query: Describe the process of photosynthesis and its importance to the Earth's
     ↪  ecosystem.
7   Response: Photosynthesis is the process by which green plants and some other
     ↪  organisms use sunlight to synthesize nutrients from carbon dioxide and water. It
     ↪  generates oxygen as a byproduct, which is vital for most life forms on Earth.

8
9   3. Title: Calculating Travel Distance
10  Query: If a car travels at 60 miles per hour for 3 hours, how far has it gone?
     ↪  Explain your calculation.
11  Response: The car has traveled 180 miles, calculated as 60 miles/hour * 3 hours.

12
13  4. Title: Utilitarianism vs Deontological Ethics
14  Query: Discuss the main differences between utilitarianism and deontological ethics.
15  Response: Utilitarianism focuses on the outcomes or consequences of actions to
     ↪  determine morality, while deontological ethics considers the actions themselves
     ↪  and the adherence to duties or rules as the basis for morality.

16
17  5. Title: Advancements in Quantum Computing
18  Query: What are the key advancements in quantum computing over the last decade?
```

19  Response: Key advancements include the development of quantum supremacy, error
    ↪  correction, and the creation of more stable qubits, enhancing computing power
    ↪  and reliability.

20

21  6. Title: Wedding Traditions in India
22  Query: Compare the wedding traditions of Northern and Southern India.
23  Response: Northern Indian weddings often feature elaborate rituals like Sangeet and
    ↪  Mehendi, while Southern Indian weddings are marked by rituals like Kashi Yatra
    ↪  and Oonjal. Both have vibrant traditions but differ in cultural practices and
    ↪  attire.

24

25  7. Title: Deforestation in the Amazon
26  Query: What are the primary causes of the Amazon rainforest's deforestation and what
    ↪  measures are being taken to address it?
27  Response: Primary causes include agriculture, logging, and infrastructure
    ↪  development. Measures to address this include enforcement of laws, satellite
    ↪  monitoring, and international cooperation on sustainable practices.

28

29  8. Title: Theme of Ambition in Macbeth
30  Query: Analyze the theme of ambition in Shakespeare's 'Macbeth'.
31  Response: Ambition in 'Macbeth' serves as both a driving force and a tragic flaw for
    ↪  the characters, particularly Macbeth, leading to his rise and eventual downfall
    ↪  as he succumbs to the ambition spurred by the prophecy and his wife's
    ↪  encouragement.

32

33  9. Title: Global Impact of Renewable Energy
34  Query: Discuss the global impact of renewable energy sources on climate change.
35  Response: Renewable energy sources like solar and wind have a significant impact on
    ↪  mitigating climate change by reducing dependence on fossil fuels, decreasing
    ↪  greenhouse gas emissions, and promoting sustainability. Countries adopting
    ↪  renewable energy contribute to a global reduction in carbon footprints, which
    ↪  can help meet the goals set by international climate agreements like the Paris
    ↪  Agreement.

36

37  10. Title: The Role of Artificial Intelligence in Healthcare
38  Query: Explain how artificial intelligence is transforming healthcare.
39  Response: Artificial intelligence in healthcare is transforming the industry by
    ↪  enhancing diagnostic accuracy, improving treatment personalization, and
    ↪  optimizing operational efficiencies. AI applications include predictive
    ↪  analytics for patient management, automated imaging and diagnostics, and
    ↪  robot-assisted surgeries, leading to faster, more accurate patient care and
    ↪  reduced healthcare costs.

40

41  11. Title: Cultural Significance of Food in Japan
42  Query: Describe the cultural significance of food in Japan and how it reflects
    ↪  Japanese society.
43  Response: Food in Japan is deeply intertwined with the nation's culture, reflecting
    ↪  aspects of beauty, seasonality, and regional diversity. Traditional dishes like
    ↪  sushi and bento embody aesthetic principles and social customs, such as respect
    ↪  for nature and meticulous attention to detail. Food rituals, such as tea
    ↪  ceremonies, also highlight the importance of mindfulness and harmony in Japanese
    ↪  society.

44

45  12. Title: Economic Effects of Globalization
46  Query: Analyze the economic effects of globalization on developing countries.
47  Response: Globalization has both positive and negative economic effects on
    ↪  developing countries. On the positive side, it allows access to international
    ↪  markets, increases capital inflow, and promotes technology transfer, leading to
    ↪  job creation and economic growth. However, it can also lead to economic
    ↪  dependency, cultural homogenization, and the potential exploitation of local
    ↪  resources and labor, which might exacerbate inequalities and social tensions.

