# OpenReview forum: "Synthesize, Partition, then Adapt: Eliciting Diverse Samples from Foundation Models"
_NeurIPS.cc/2024/Conference — NeurIPS 2024 poster_

### Official Review · Reviewer_4Daa · 2024-07-07

**Soundness:** 3
**Presentation:** 3
**Contribution:** 2
**Rating:** 4
**Confidence:** 3

**Summary:**

This paper proposes a method to generated diverse responses for the language model. Given a synthetic finetuning dataset and test dataset, they calculate the importance of the finetuning data on each test data using methods such as influence function, and then partition the dataset into several smaller ones. Finally, they finetune several lora weights based on different partition datasets.

**Strengths:**

The authors conducted experiments to verify that the proposed SPA based on influence function are validated to improve the diversity among baselines.

**Weaknesses:**

It involves finetuning, and need to finetune several lora weights. This would cause difficulty on parallel generation because we need to store multiple checkpoints into memory.

**Questions:**

How to find the optimal number of partition in practice? I suppose there should be a sweet point.

Can authors report the actual time on calculating influence function and training the lora models for the experiments in this paper?

If you need multiple lora weights to generate diverse responses, are you able to still generate them in parallel without demanding memory requirement?

**Limitations:**

.

---

> ### Author Rebuttal · Authors · 2024-08-06
>
> Thank you for your insightful review. We'd like to address your main concerns.
> > Parallel Generation and Memory Efficiency
>
> Recent advances in serving multiple LoRA adaptations in parallel significantly mitigate the need to store multiple full checkpoints. Notably, the S-LoRA system [1] demonstrates that it's possible to efficiently serve thousands of LoRA adapters with minimal overhead, using techniques like Unified Paging to manage memory efficiently. Furthermore, FLoRA [2] allows each input example in a minibatch to be associated with unique low-rank adaptation weights, enabling efficient batching of multiple LoRA adapters. In these systems, we only need to store the LoRA weights in memory, which are often low-rank, significantly reducing memory requirements. These advancements suggest that SPA can be implemented without significant memory constraints or parallel generation difficulties. As the field continues to progress, we anticipate even more efficient solutions for managing multiple LoRA adapters.
>
> [1]: Sheng, Y., Cao, S., Li, D., Hooper, C., Lee, N., Yang, S., Chou, C., Zhu, B., Zheng, L., Keutzer, K., Gonzalez, J.E., & Stoica, I. (2023). S-LoRA: Serving Thousands of Concurrent LoRA Adapters. MLSys 2024
>
> [2]: Wen, Y., & Chaudhuri, S. (2023). Batched Low-Rank Adaptation of Foundation Models. ICLR 2024
> > Best number of partitions
>
> Determining the optimal number of partitions is indeed an important consideration. This highly depends on the size of the synthetic dataset, with larger datasets typically inducing a higher optimal number of partitions. We conducted an ablation study (Fig. 6 in our paper) examining the effect of the number of partitions on diversity. We found that the diversity score remains relatively stable as the number of adaptations increases from 8 to 12, suggesting that 8 partitions is roughly the sweet spot for the synthetic dataset we used (75k data points). We also observed improvements in diversity when increasing the number of partitions from 4 to 8, though this data was not included in Fig. 6.
> > Computational time
>
> We provided a rough complexity in the limitation section. To offer more details: One epoch of fine-tuning on the 75k synthetic data points takes approximately 8 hours using 2 A40 GPUs. We trained the LoRA weights for 4 epochs. Influence function calculation costs roughly 1 epoch of the training time, which is approximately 8 hours. We want to highlight that the main computational cost occurs during the offline partitioning and adaptation phases.
> Future work could explore more efficient data attribution methods, such as TRAK [1] and K-FAC [2], which have the potential to substantially reduce this overhead while maintaining the benefits of our approach.
>
> We appreciate your feedback and hope our clarifications have addressed your concerns about parallel generation and memory requirement. If you have any remaining concerns, please let us know. We're grateful for the opportunity to further explain our work.
>
>
> [1]: Park, S.M., Georgiev, K., Ilyas, A., Leclerc, G., & Madry, A. (2023). TRAK: Attributing Model Behavior at Scale. International Conference on Machine Learning. ICML 2023
>
> [2]: Grosse, R.B., Bae, J., Anil, C., Elhage, N., Tamkin, A., Tajdini, A., Steiner, B., Li, D., Durmus, E., Perez, E., Hubinger, E., Lukovsiut.e, K., Nguyen, K., Joseph, N., McCandlish, S., Kaplan, J., & Bowman, S. (2023). Studying Large Language Model Generalization with Influence Functions.

---

> > ### Comment · Reviewer_4Daa · 2024-08-12
> >
> > Thank you for your rebuttal! I'm still a bit uncertain on "These advancements suggest that SPA can be implemented without significant memory constraints or parallel generation difficulties. As the field continues to progress, we anticipate even more efficient solutions for managing multiple LoRA adapters." Are you able to implement the combination with some of these methods you mentioned and show some results?

---

> ### Author Response · Authors · 2024-08-12
>
> Thank you for your follow-up question. We conducted two simplified scenarios using S-LoRA codebase and vLLM to compare sampling from multiple LoRA components versus a single model.
> Our setup uses a synthetic dataset in S-LoRA repo with 600 prompts and responses ranging from 8 to 512 tokens respectively.
> For each prompt, the model generates tokens until the output length matched the response's length. We used Llama-13b with LoRA rank 16, leading to a 10% memory overhead for storing 8 LoRA adapters compared to a single model. When running the serving experiment, the requests are coming with a request rate of 2 per second.
>
> In the first experiment, we simulated a scenario where multiple GPUs were used for serving. Here, the goal was to generate 8 diverse samples per request. We used 8 A100-40G GPUs. For the single model setup, incoming requests (say x and y) were expanded into batches of 8 identical prompts, $[x_1, x_2, ... , x_8]$ and $[y_1,y_2,...,y_8]$ and handled by vLLM’s continuous batching and routing. Usually, it will distribute $[x_1, x_2]$ to GPU-1 job queue, $[x_3, x_4]$ to GPU-2 job queue and so on.
>
> For the SPA model, the batch becomes $[x_i, y_i, LoRA_i]$, where i ranges from 1 to 8.  The router then distributed these to different GPU job queues, with each GPU handling a unique prompt-LoRA pair.
> This also gives us 8 samples per request. The results showed that this leads to a minor 3% overhead in both throughput (tokens / sec) and latency (secs / output token), primarily due to the additional step of distributing the LoRA adapters. This approach ensures parallel generation but requires multiple GPUs. However, it is a common practice to use multiple GPUs to serve real-world requests.
>
> | 2 req/s   | tokens / sec | secs / output token |
> |----------------|----------|------------|
>  |  Single | 473   |  0.035   |
> |  SPA  |  460  |  0.036     |
>
>
> In the second experiment, we explored a single GPU scenario using S-LoRA. This setup also stores all LoRA weights in memory, leading to a similar 10% memory overhead as in the first scenario. In this experiment, we only generated one sample per request for simplicity. This means that for the single model, the metrics should be roughly the same as those obtained when sampling 8 outputs using 8 GPUs in the above scenario. In the SPA setup, each request was coupled with a random LoRA adapter, which is handled by the S-Lora library. The result below shows that this configuration leads to a ~10% overhead in throughput and latency.
>
> | 2 req/s   | tokens / sec | secs / output token |
> |----------------|----------|------------|
>  |  Single | 480   |  0.035     |
>  | SPA  |  433 |  0.040     |
>
>
> The first result indicates that SPA can be implemented efficiently in a multi-GPU setup with almost no performance lost. Even in a single GPU scenario, leveraging S-LoRA allows us to sample from multiple LoRA adapters without significant trade-off. This is also an active research field, we could anticipate even more efficient solutions for sampling from multiple LoRA adapters with some future works. We hope these results help address your concerns.

---

> ### Comment · Reviewer_4Daa · 2024-08-13
>
> Thank you! I am afraid that I would still keep my score because in a high level, I am not quite in favor of the idea of having multiple lora weights.

---

> > ### Author Response · Authors · 2024-08-14
> >
> > Thank you for your engagement throughout this process. Our approach assumes LLMs undergo post-training with synthetic data, a common practice (e.g., Llama 3's iterative post-training). Given the constraints of this paradigm and motivated by diminishing returns of increasingly many synthetic data, sampling from multiple distributions using LoRA is our strategy to outperform a single model tuned on the entire synthetic dataset. This strategy aims to generate diverse, high-quality responses while addressing the challenge of diminishing returns in LLM post-training with synthetic data.
> >
> > Although you may not be fully convinced, we hope that our paper and the additional experiments we've provided will contribute more insights to the later discussion.

---

### Official Review · Reviewer_9mQS · 2024-07-12

**Soundness:** 3
**Presentation:** 4
**Contribution:** 3
**Rating:** 6
**Confidence:** 3

**Summary:**

This paper presents a framework for eliciting diverse outputs from language models while maintaining quality/accuracy. The framework consists of: first partitioning a (synthetic) dataset of supervised instruction tuning data, then training parameter efficient model adapters on each partition, and finally at inference time sampling from each of the the models trained in the prior step.

Within this framework, the paper contributes empirical evaluations of 2 methods (influence functions and token overlap) for partitioning the instruction tuning datasets and measures the effects on both code completion and natural language generation benchmarks. They find that both methods generate more diverse output over random partitioning (i.e. an ensemble of models trained on random subsets of the data), suggesting that the method used to partition is important.

The authors contrast their method with temperature based sampling approaches to generating diverse data from an LLM.

**Strengths:**

*   This paper recognizes certain diminishing returns of scaling up dataset size and instead finds alternative uses of large datasets to generate diverse outputs. This is neat approach to thinking about other ways to use "scaled-up" data.
*   Demonstrates that the data partitioning approach does result in significant differences wrt diversity at lower temperatures. In particular the authors demonstrate that model ensembles of randomly partitioned data do not generate as diverse outputs as ones trained on partitions generated more principled approaches.
*   The paper is overall well written and easy to read.

**Weaknesses:**

*    Evaluation on natural language (non-code) tasks lacks a quality/accuracy metric for the proposed method (Fig 5). While it is great to see that diversity is increased, the authors do state that maintaining quality is important but it is unclear if that is the case for these tasks. It would be helpful to see "standard" eval metrics for some or all of MMLU, BBH, GPQA and WinoGrande using the methods proposed.
*   More clarity in the tradeoff the proposed methods make vs high temperature sampling would be helpful. At least for the code models, it appears that as temperature increases, the diversity of the single model might converge to that exibited by SPA. If that is the case some discussion of the tradeoffs involved, (e.g. benefits of being able to sample at low temperature) would strengthen the paper.
*   It is difficult to know how sensitive the results are to the diversity of queries used to generate the partitioning matrix? It would be helpful to understand the performance of a partitioning method that is based on the data but doesn't use human selected test queries (e.g. randomly select 12 prompts from HumanEval and then partition based on that using influence and token overlap approaches). This is not however a fatal flaw by any means.

**Questions:**

Experiment Design

*   How were the 12 queries for each task generated? Could the authors shed any light on what they think important properties of this query set are?
*   What are the generation params for Section 5.3? (is it all greedy sampling)
*   Also in Section 5.3, why not compare with a single model at high temperature (Similar to the "Single" condition used for the code generation tasks)?

*   Results
*   How do the charts in figure 4 look like as temperature increases even further? Why did you stop at 0.5? The CodeLLama paper reports their main results at temperate=0.8 so it would be useful for readers to see if and where the methods converge in terms of diversity.
*   Did you try using fewer than 8 model adapters? While there doesn't seem to be any much improvement observed in using more than 8 adapters in this setting, do you have a sense of how few adapters one could use?


*   Metrics

*   For the average KL divergence metric could you explain how you go from token level probability distributions for sequence level probability distribution for Pi and Pj? Or do you mean something else, I'm mostly trying to understand how you handle sequences of different lengths in the computation.
*   For the sample diversity/Diversity score metric what value of K (line 273) is used in the experiments? And is the total number of samples generated in each condition the same?
*   In figure 4 how many samples are generated for each method to calculate pass\@k? And in the case of the SPA methods how is that distributed across the various adapters? My understanding is that there are 8 adapters in this experiment.
*   Do the authors have any thoughts on why pass@1 isn't higher for SPA compared to single model?

**Limitations:**

Yes. Though the paper could also better acknowledge the extra computational cost at inference time (compared to just sampling multiple times at high temperature).

---

> ### Author Rebuttal · Authors · 2024-08-06
>
> Thank you for your insightful review. We'd like to address your main concerns.
> > Accuracy on NL tasks
>
> The accuracy of SPA methods (influence-based, lexical-based, and random) all achieve similar accuracy to the single model, reaching the same conclusion as the pass@1 metric in Fig. 4a. We didn't observe improved pass@k with increased diversity on these tasks, likely because most are multiple-choice questions. For diversity measurement, we asked models to continue generating text even after producing an answer choice. We omitted accuracy results as they didn't provide new insights beyond the code generation tasks. However, we acknowledge that including this data would have provided a more complete picture and will add it to the revised paper.
> > Trade-off in high temperature sampling, temperature plot stops at 0.5
>
> You're correct that at higher temperatures, the diversity of single models converges to SPA models. In practice, low temperature or even greedy sampling is often preferred when average sample quality is crucial. As shown in Fig. 4, high temperature leads to a decrease in the pass@1 metric, indicating a drop in sample quality. High temperature sampling may deviate from the learned distribution and produce hallucination or less coherent outputs [1]. We briefly discussed the disadvantages of high temperature in the introduction and will expand it in the revision.
>
> We limited our plots to temperatures up to 0.5 as SPA's primary goal is to maintain average sample quality (measured by pass@1) while achieving diversity. Fig. 4a shows that temperature 0.5 already leads to a significant drop in pass@1. However, we acknowledge the importance of understanding where methods converge in terms of diversity. Our preliminary experiments indicate convergence around temperature 0.8. We will include these higher temperature results in our revision.
> > Number of adapters
>
> The optimal number of adapters highly depends on the size of the synthetic dataset, with larger datasets typically inducing a higher optimal number of adapters. We observed improvements in diversity when increasing the number of adapters from 4 to 8, though this data was not included in Fig. 6,  suggesting that 8 adapters is roughly the sweet spot for the synthetic dataset we used (75k data points).
> > KL divergence, hyper-parameters, distribution over LoRA adapters
>
> We measured the KL divergence at the first decoding step, which is at the token level. This measures the difference in states across different adaptations after seeing the same prompts. This is not applicable to the single model approach.
>
> For each method (when temperature > 0), we sampled a total of 120 samples to calculate pass@1, pass@k, and diversity score (K=5). This larger sample size helps reduce metric variance. For SPA methods, we distributed samples evenly among adapters (15 samples per adapter). Future work could explore weighted sampling in SPA. The exception is greedy sampling (temp=0 in Fig. 4 and Sec. 5.3), where we only get one sample per adapter, leading to a total of 8 samples.
> > Why pass@1 is not higher, how to select queries.
>
> SPA is primarily designed to improve diversity, which doesn't necessarily improve average sample quality. Hence, it shows similar pass@1 performance to the single model.
>
> Our principle was to create a diverse query set. We used few-shot prompting with ChatGPT to generate ~20 queries, then manually selected 8 covering a wide range of topics.
> > Single model at higher temp in Sec. 5.3
>
> We didn't include comparisons with single models at higher temperature because, as shown in Fig. 4a, higher temperatures lead to sample quality drops. However, we acknowledge that including this comparison could provide valuable insights. In our revision, we will add this comparison and discuss the trade-offs.
>
> We appreciate your feedback and hope this clarifies your concern. Please let us know if there is any remaining concerns.
>
>
> [1]: Lee, M. (2023). A Mathematical Investigation of Hallucination and Creativity in GPT Models. Mathematics.

---

> > ### Comment · Reviewer_9mQS · 2024-08-12
> >
> > I thank the authors for their response and clarifications. I overall still think this is an interesting paper!
> >
> > It might be helpful to post the accuracy scores for the NL tasks that you mentioned in your rebuttal here. If accepted readers may also be interested in the distribution of accuracy across the various adapters.

---

> ### Author Response · Authors · 2024-08-13
>
> Thank you for your positive feedback. Here are the accuracy results:
>
> | Tasks   | BBH | GPQA | MMLU | WinoGrande |
> |----------------|----------|------------|------------|------------|
> |  Single |  46.66  |    4.14  |   55.40   |   76.5     |
> |  SPA    |   46.48 |    3.80  |   55.26  |   76.3    |
>
>
> SPA maintains comparable accuracy to the single model. The GPQA evaluation has high variance; even small changes in the prompt can lead to big differences in accuracy. Despite this, we still included GPQA because it provides a good source for evaluating diversity due to its broad range of topics. We will include the distribution in the next revision as you suggested.
>
> We've also added new experiments in our reply to reviewer 4Daa addressing concerns about parallel generation and memory overhead. We greatly appreciate your thoughtful feedback and hope we have addressed your concerns. We're keen on your support for our submission.

---

### Official Review · Reviewer_j8SA · 2024-07-13

**Soundness:** 1
**Presentation:** 2
**Contribution:** 1
**Rating:** 4
**Confidence:** 5

**Summary:**

The paper introduces Synthesize-Partition-Adapt (SPA), a framework designed to generate diverse and high-quality responses from foundation models. SPA uses synthetic data and data attribution methods to partition data into subsets, training multiple model adaptations for these subsets.

**Strengths:**

I believe this paper is studying an important problem. The diversity problem in LLM has received attention recently.

**Weaknesses:**

- The motivation of three phases: synthesize, partition, then adapt is not clearly discussed (See my questions). The authors spend more time to describe what they are doing but not why they are doing so or why these specific designs would lead to desired outcome, which is diverse sampling technique.

- The experiments only compare with two simple baselines ablated from the proposed methods, omitting existing baselines in the literature.

- The paper omits most of the existing works in the literature regarding diverse sampling for LLMs and only considers simple temperature sampling techniques.

**Questions:**

- The motivation of three phases: synthesize, partition, then adapt is not clearly discussed. How to ensure that the existing dataset is diverse?

- The framework heavily relies on the existing synthetic dataset where its diversity is already a concern. As the authors mentioned, the problem of generating diverse synthetic dataset is still a problem with current generative models. Would finetuning on these synthetic datasets destroy the generalizability of the models?

- The method relies on $M$ questions requiring various expertise knowledge to compute influence matrix for partitioning. Does those M questions need to be manually chosen?

- Why partitioning the datasets using the importance scores and then finetune different LoRAs with K partitions would lead to K diverse LoRAs components? The connection is vague to me.

- The method relies on a predefined number of partitions K (or finetuned LoRA components) when training. What if during the inference time, we would generate a lot more than K diverse responses? Does each finetuned model still suffer from diversity problem?

- During inference, LoRA components are sampled randomly, does it affect the quality of generated responses? What if some components being biased or learn some skills while losing other skills? It is not clear that, after partition and finetuning, which LoRA learns which expertise. Can each component has a weight depending on the question?

- The paper is missing most of the important baselines and citations, I listed some here for reference:

    - KL-Divergence Guided Temperature Sampling

    - Diversity of Thought Improves Reasoning Abilities of LLMs

    - Large Language Models as In-context AI Generators for Quality-Diversity

    - Controlled Decoding from Language Models

    - Quality-Diversity through AI Feedback

    - Language Model Decoding as Direct Metrics Optimization

**Limitations:**

yes

---

> ### Author Rebuttal · Authors · 2024-08-07
>
> Thank you for your insightful review. We'd like to address your main concerns.
> > Motivation of why doing SPA and why doing so leads to diverse samples. Role of synthetic data and the generalizability of it.
>
> We would first like to clarify that LLMs must typically go through synthetic data tuning before deployment, whether for domain adaptation, value alignment, or distilling knowledge from a stronger model. SPA is designed to **utilize** this existing synthetic data used in LLM post-training stages (or mid-training stage). Operating within this existing paradigm, SPA addresses the inefficiency of tuning a single model on **increasingly larger** synthetic datasets due to diminishing returns.
>
>
> The first motivation, as stated in Section 3, is to turn this potential inefficiency into an advantage for diversity from the multi-model perspective. The second motivation is sampling from multiple distinct distributions (i.e., different model adaptations) naturally yields more diverse outputs compared to sampling multiple times from a single distribution (i.e., a single model).
> This is why SPA leads to diverse samples by creating multiple model adaptations, each trained on a different partition of the synthetic data. Importantly, due to diminishing returns, training on data partitions doesn't significantly affect accuracy, as shown in Sec. 3.
>
> Furthermore, our partitioning strategy, especially when using influence functions, creates subsets of data that induce distinct model skills. This further enhances the diversity of the resulting adaptations (compared to the random and lexical partitions).
>
> Regarding generalizability, since SPA uses data already part of the post-training process, it doesn’t introduce new generalization risks. While our method benefits from more diverse data, it's designed to improve diversity even when working with less diverse synthetic datasets, by being able to sample from multiple distinct distributions.
> > More comparisons and related works
>
> Thank you for this feedback.  For the additional comparisons, please see the general response. We didn't compare to QD search methods because SPA and QD methods operate on fundamentally different principles. SPA operates at the model level, creating multiple models during an offline process. The main computational cost occurs during the offline partitioning and adaptation phases. This allows for quick, parallel diverse sampling at inference time. In contrast, QD search operates at the generation level, iteratively going through mutation, evaluation, and refinement cycles. It introduces overhead during sampling, making it less suitable for real-time applications. We will also expand our literature review to include all the suggested works and other relevant studies.
> > Connection between partitioning and diverse LoRA components
>
> The influence function captures how training examples affect model predictions on specific test queries. Our partitioning strategy groups examples with similar influence patterns, resulting in K clusters. Each cluster consists of synthetic data that most strongly influences a particular test query (detailed at line 257). When fine-tuning LoRA on these individual clusters, each component focuses on the skills emphasized by its associated test query.
>
> Importantly, we don't require extremely diverse LoRA components because the diversity primarily stems from sampling across multiple distinct distributions (motivation #2). Even with random partitioning, this strategy ensures that these distributions are sufficiently different to generate diverse outputs when sampled collectively.
> > Generating more than K diverse responses
>
> While we train K adapters, SPA is not limited to generating only K diverse responses. Each adapter can produce multiple outputs, especially when combined with other sampling techniques (e.g., temperature sampling). In our experiments, for temperatures > 0, we generated a total of 120 samples to calculate pass@1, pass@k, and diversity scores. For SPA, this meant sampling 15 samples from each of the 8 adapters. In contrast, for the single model baseline, all 120 samples were drawn from the same distribution. This sampling strategy highlights a key advantage of SPA: even when generating many more than K samples, we're still drawing from multiple distinct distributions. This inherently promotes diversity compared to repeatedly sampling from a single distribution.
> Moreover, SPA is complementary to other diversity-enhancing methods.
> > Quality and bias in randomly sampled LoRA components
>
> Our experiments show that random sampling of LoRA components during inference doesn't significantly affect response quality, as evidenced by our pass@1 results (also because of motivation # 1). Our partitioning strategy aims to create balanced subsets that cover different aspects. The additional comparisons in the general response showed that SPA is less biased than diversity of thought. However, we acknowledge that some components might specialize more in certain areas. In practice, a held-out dataset can be used to identify and remove overly biased components before deployment. Future research could explore weighted component selection, using a similar routing strategy in MOE.
> > Selection of M queries
>
> The M questions don't necessarily need to be manually chosen. In our experiments, we used a semi-automated approach: we few-shot prompted ChatGPT to generate ~20 diverse queries and then manually selected 8 of them to cover a wide range of topics (Appendix C).
>
> We appreciate your feedback and hope our clarifications have addressed your concerns about motivation and the empirical results.  If you have any remaining concerns, please let us know. We're grateful for the opportunity to further explain our work.

---

> > ### Comment · Reviewer_j8SA · 2024-08-10
> > **Thank you!**
> >
> > Thank the authors for the rebuttal and conducting additional experiments. For this reason, I increased my score, however, I still tend to rejection for this paper since I'm not really being convinced by diverse sampling process only via multiple LoRA components. Each LoRA component still suffers from the diversity sampling problem since  the training procedure for each LoRA component is still the same.
> >
> > Furthermore, calibrating the quality-diversity trade-off typically depends on each use case. Predefining K before finetuning makes it's difficult for the users to calibrate this trade-off, thus limiting its use case in practice. In those cases, we then need to come back to calibrate this trade-off by temperature, which, as the authors argued, has drawbacks. This also relates to my concerns regarding random choices of LoRA components during sampling. What if after finetuning, we want to generate the most probable, high-quality answer?

---

> ### Author Response · Authors · 2024-08-13
>
> We sincerely appreciate your reconsideration and the increased score. We'd like to take the opportunity to address your remaining concerns:
>
> > The same training procedure for each LoRA component leads to less diversity
>
> Our approach operates under the assumption that LLMs undergo post-training using synthetic data, which is a common practice (e.g., iterative post-training in Llama 3 utilizes synthetic data in each round). Under this assumption, we have limited flexibility to modify the training procedure.
> Given these constraints, and motivated by diminishing returns, sampling from multiple distributions using LoRA components is our strategy to outperform a single model tuned on the entire synthetic dataset. While each LoRA component on its own may face diversity challenges like you mentioned, it won't be worse than the single model scenario.
>
> Moreover, if modifying the training procedure is allowed, we could incorporate the mutual information as one of the objectives when optimizing LoRAs, this will encourage these LoRA components to be as diverse as possible.
>
>
> > Quality-diversity trade-off
>
> We agree that calibrating this trade-off may be challenging in practice. On the other hand, SPA offers an improved Pareto frontier, improving diversity even at low temperatures. Users can now achieve good diversity at low temperatures (even when greedy), which was previously difficult. Thus, while SPA doesn't eliminate the trade-off, it provides users with more options.
>
>
> > Predefined K and component quality, get the most probable sample
>
> We want to clarify that K is not arbitrarily selected but depends on the synthetic dataset size and the strength of diminishing returns. As shown in Fig 4a, $\sum_i^K \text{Accuracy}(\text{model}_{\phi_i}) / K$, roughly equals the accuracy of a single model $\phi$, fine-tuned on the entire synthetic dataset. The average quality of LoRA components is roughly the same as the single model baseline.
>
>
> For cases requiring the most probable answer (only one sample is required), we can conduct an offline evaluation on a held-out dataset before deployment, then sample from the best-performing LoRA component.
>
> We hope our clarifications can further address your concerns.

---

### Official Review · Reviewer_BWst · 2024-07-14

**Soundness:** 3
**Presentation:** 3
**Contribution:** 4
**Rating:** 8
**Confidence:** 4

**Summary:**

This paper aims to improve the diversity of generations of LLMs. Current methods are not great because when they offer more diversity, they lose in quality (for instance worse performance). This is for instance the case of temperature sampling. In this paper, they propose synthetise partition and adapt which consists in taking a synthetic dataset, partitioning it using influence function for instance and a few high quality test examplars, train a multiple models on each of these subsets and then eventually, use this collection of models to do sampling. They show that thanks to their approach, they are able to get better diversity while maintaining good performance on code and MMLU.

**Strengths:**

Overall, I like this paper. I think that the question addressed by the authors is central and to me one of the biggest issues with current LLMs is their lack of their generations diversity. To my knowledge, this paper is one of the first ones that proposes a method to alleviate this issue and I hope many other works will follow. The paper is well presented, easy to read and the results are clear.

**Weaknesses:**

I want to raise a few points regarding the approach:

- **In-context approach?**: I think that one criticism could be that he method may seem a bit computationally heavy (because of the data clustering step + training many models), I know that one of the most commonly used methods in practice to ensure diversity is to select different text chunks and use them to seed the prompt and get a different generation. This was used for instance in the Phi and cosmopedia approaches to generate synthetic textbooks [1]. In other words, do you think that the finetuning step is necessary? Wouldn't putting one of the clusters of data (that you create with data attribution) **In-context** be enough?


- **Role of the rank of LoRA**: I imagine that the rank of LoRA may play a role here? Intuitively, if the rank is very small, all the models will have similar representations even after fine-tuning. As you increase the rank of LoRa, you may increase the diversity of your generations. Is it correct?

- **Diversity measures**: The diversity measures proposed by the authors are legit. However, maybe another interesting one would be to train a new model on datasets that they have generated with their approach. If the newly trained model gets a better performance, I think this would be the best proof for diversity.

- **Increasing the number of clusters leads to better diversity + performance?**: the current paper just considers the case where there are 8 clusters in the data attribution phase. One experiment that would have been nice is also to understand in the case of code, how pass@5 improves as the number of clusters increase ?

- **Some improvements that may be done for the introduction**: I think it is a shame that the method is not fully presented in the introduction. Maybe it would be good to summarize it in a few words? Besides, I believe that it would be nicer if Figure 3 is in page 2 as a main figure.



[1] Gunasekar, S., Zhang, Y., Aneja, J., Mendes, C. C. T., Del Giorno, A., Gopi, S., ... & Li, Y. (2023). Textbooks are all you need. arXiv preprint arXiv:2306.11644.

**Questions:**

I mentioned my questions in the weaknesses seciton.

**Limitations:**

The authors mentioned the limitations of their work and they look sound to me.

---

> ### Author Rebuttal · Authors · 2024-08-06
>
> Thank you for your insightful review. We'd like to address your main concerns.
> > In-context v.s. Fine-tuning
>
> We can formulate the difference as sampling from $P_{\phi_1} (y|x)$ and $P_{\phi_1}(y|x')$ for in-context versus $P_{\phi_1}(y|x)$ and $P_{\phi_2}(y|x)$ for our approach ($\phi$ denotes model parameters). We argue that sampling from multiple distinct distributions (SPA) produces more diversity than modifying the prompt. In the general response, we conducted additional experiments comparing SPA with Diversity of Thought, an in-context approach. The results show SPA's significant advantage. Moreover, one could potentially combine them by sampling from $P_{\phi_1}(y|x)$ and $P_{\phi_2}(y|x')$, further enhancing diversity.
> Finally, we assume the LLM must go through the synthetic data tuning before deployment, whether for domain adaptation, value alignment, or distilling knowledge from a stronger model. Given the diminishing returns observed with increasing synthetic data, the motivation for SPA remains strong.
> > Role of LoRA rank
>
> We agree that your intuition about LoRA rank is correct. A smaller rank would lead to more similar representations, potentially reducing diversity, while a larger rank might enhance it. We used a fixed rank for consistency in our experiments, but exploring different ranks is an interesting direction for future work. Yet, the benefit of sampling from multiple distinct distributions remains valid regardless of the rank.
> > Number of adapters
>
> The optimal number of adapters highly depends on the size of the synthetic dataset, with larger datasets typically inducing a higher optimal number of adapters. We observed improvements in diversity when increasing the number of adapters from 4 to 8, though this data was not included in Fig. 6, suggesting that 8 adapters is roughly the sweet spot for the synthetic dataset we used (75k data points).
> > Further improvements
>
> Thank you for your value suggestions for new diversity measures and how to improve the intro section. Regarding the new diversity measure, we'd appreciate clarification on training a new model on generated datasets, as our single model adaptation baseline is already trained on the entire synthetic data. We’ll summarize the method more concisely in the intro and move Fig. 3 ahead in our revision.
>
> We appreciate your positive feedback and hope this addresses your concerns. We kindly ask for your support of our paper.

---

> > ### Comment · Reviewer_BWst · 2024-08-09
> >
> > I thank the authors for responding to my questions. After reading the other reviews, I still believe that the paper is meaningful, has an interesting contribution and may benefit the community. I maintain my score.

---

> ### Author Response · Authors · 2024-08-13
>
> Thank you for your positive feedback. We've also added new experiments in our reply to reviewer 4Daa addressing concerns about parallel generation and memory overhead. We're keen on your support for our submission.

---

### Author Rebuttal · Authors · 2024-08-06

We thank all reviewers for their valuable feedback. Here, we address the concern raised by reviewer j8SA regarding limited comparisons.

Our initial comparisons focused on ablations as SPA represents a novel multi-model adaptation method addressing diminishing returns of abundant synthetic data. Unlike most methods that modify the sampling process, SPA operates at the model level, making it complementary to existing methods.
To provide more context, we've conducted additional experiments comparing SPA with KL-divergence guided temperature sampling [1] and Diversity of Thought [2].

For KL-guided temperature sampling, following [1], we maintained two parallel decoding sequences: one with full information and another identical but without the docstring (mainly test cases). At each decoding step, we calculated the KL divergence between the logits of these two sequences' forward passes, using it to rescale the temperature. The newly sampled token was then added to both sequences.

We iterated hyperparameters $T_0 = \\{0.1, 0.2, …, 0.5, 0.8, 1.0\\}$ and $\sigma = \\{1e-3, 1e-2, 0.1, 1, 10\\}$, reporting the best runs for pass@1 and pass@5 separately.  Notably, no combination achieved both high pass@1 and pass@5 simultaneously like SPA did. The results show that KL-guided temperature scaling generates diverse samples but at the cost of average sample quality.

For the DIV-SE, following Fig.5 in the paper, we generated 8 reasoning approach templates for HumanEval problems. We only used 3 of them (Iterative, BruteForce, Greedy) in the final evaluation due to poor pass@1 performance of the others. This is because the reasoning templates don't generalize well to the diverse coding challenges in HumanEval though they work well in the math domain. The result shows that SPA is better in terms of all metrics when sampled at different temperatures.

These comparisons further show SPA's advantage in sampling from multiple distinct distributions.
| HumanEval   | pass@1 | pass@5 | Diversity |
|----------------|----------|------------|---------------|
|  DIV-SE temp 0.1  | 46.78   |  58.70     |   0.72     |
| Influence-SPA temp 0.1 | 49.80  |  69.59     |  0.85    |
|
|  DIV-SE temp 0.2  | 45.61   |  61.70     |   0.79     |
| Influence-SPA temp 0.2 | 48.74  |  69.82     |  0.86    |
|
|  DIV-SE temp 0.3  | 44.57   |  62.01     |   0.84     |
| Influence-SPA temp 0.3 | 47.64  |  70.89     |  0.88    |
|
|  DIV-SE temp 0.4  | 43.63   |  63.34     |   0.88     |
| Influence-SPA temp 0.4 | 46.97  |  71.91     |  0.91    |
|
|  DIV-SE temp 0.5  | 42.01   |  64.36     |   0.91     |
| Influence-SPA temp 0.5 | 46.82  |  72.49     |  0.92    |
|
| KL-guided temp 0.1 | 49.24 | 60.84 |  0.54 |
| KL-guided temp 0.8 | 42.17 | 66.63 |  0.93 |



[1] Chang, C., Reitter, D., Aksitov, R., & Sung, Y. (2023). KL-Divergence Guided Temperature Sampling. ArXiv, abs/2306.01286.


[2] Naik, R., Chandrasekaran, V., Yuksekgonul, M., Palangi, H., & Nushi, B. (2023). Diversity of Thought Improves Reasoning Abilities of LLMs. ArXiv, abs/2310.07088

---

### Decision · Program_Chairs · 2024-09-25

**Decision:**

Accept (poster)

**Comment:**

This paper addresses the critical challenge of generating diverse responses from foundation models while maintaining output quality. The proposed Synthesize-Partition-Adapt (SPA) framework offers a novel approach by leveraging synthetic data and model adaptations, distinguishing itself from methods that solely modify the sampling process. The experimental results, particularly in code generation tasks, demonstrate the effectiveness of SPA in improving diversity metrics without sacrificing performance. While concerns were raised about comparisons to existing baselines and the practicality of serving multiple LoRA weights, the authors have adequately addressed most of these issues in their rebuttal (including the benchmarks for practical serving of LoRA weights based on recent papers during the rebuttal phase). The importance of the problem and the innovative nature of the solution outweigh the remaining concerns, making this paper a valuable contribution to the field. Therefore, the paper is recommended for acceptance to NeurIPS 2024.
The authors are encouraged to consider the feedback of the reviewers in finalizing their paper, including empirical or analytical comparisons with state-of-the-art diverse sampling methods to provide a more comprehensive evaluation of SPA's performance, providing more insight into the scalability and real-world implementation of serving multiple LoRA weights, as well as considering writing improvements starting with the introduction, summarizing the method more concisely and moving key figures earlier in the paper for better reader engagement. In the future, authors might also want to explore the combination of SPA with in-context learning approaches to potentially further enhance diversity.